# Primary Aneurysmal Bone Cyst and Its Recent Treatment Options: A Comparative Review of 74 Cases

**DOI:** 10.3390/cancers13102362

**Published:** 2021-05-14

**Authors:** Nils Deventer, Martin Schulze, Georg Gosheger, Marieke de Vaal, Niklas Deventer

**Affiliations:** Department of Orthopedics and Tumororthopedics, Albert-Schweitzer-Campus 1, University Hospital Münster, 48149 Muenster, Germany; nils.deventer@ukmuenster.de (N.D.); martin.schulze@ukmuenster.de (M.S.); georg.gosheger@ukmuenster.de (G.G.); Marieke.devaal@ukmuenster.de (M.d.V.)

**Keywords:** aneurysmal bone cyst, intralesional curettage, polidocanol injection, polidocanol instillation

## Abstract

**Simple Summary:**

This single-center study is a retrospective review of 74 patients with primary aneurysmal bone cysts (ABCs). It aims to compare the three most common treatment options—intralesional curettage, the percutaneous instillation of polidocanol and en bloc resection. It is the first study comparing these three treatment options using MR scans for the evaluation of the treatment success of instillation therapy and demonstrates the superiority of MRI scans compared to conventional radiographs for cyst volume measurement. The study confirms the efficacy of polidocanol instillations for primary ABCs and justifies it as standard treatment option. However, in this study several instillations were found to be necessary to achieve complete healing or at least stable disease. In a relevant number of cases a conversion to a surgical treatment was necessary. Thus, in this study we relativize the so-far highly positive treatment results reported for instillations in the literature, and rate them skeptically.

**Abstract:**

(1) Background: An aneurysmal bone cyst (ABC) is a benign, locally aggressive tumor. Different treatment modalities are described in the literature i.e., en bloc resection, intralesional curettage and percutaneous sclerotherapy. (2) Methods: This single-center study is a review of 74 patients with primary ABCs who underwent a surgical treatment or polidocanol instillation. Cyst volume measurements using MRI and conventional radiographs are compared. (3) Results: The mean pre-interventional MRI-based cyst volume was 44.07 cm^3^ and the mean radiographic volume was 27.27 cm^3^. The recurrence rate after intralesional curettage with the need for further treatment was 38.2% (13/34). The instillation of polidocanol showed a significant reduction of the initial cyst volume (*p* < 0.001) but a persistent disease occurred in 29/32 cases (90.6%). In 10 of these 29 cases (34.5%) further treatment was necessary. After en bloc resection (eight cases) a local recurrence occurred in two cases (25%), in one case with the need for further treatment. (4) Conclusions: MRI scans are superior to biplanar radiographs in the examination of ABCs. Sequential percutaneous instillations of polidocanol are equally effective in the therapy of primary ABCs compared to intralesional curettage. However, several instillations have to be expected. In a considerable number of cases, a conversion to intralesional curettage or en bloc resection may be necessary.

## 1. Introduction

Aneurysmal bone cysts (ABCs) are benign, locally destructive growing bone tumors, which were first described by Jaffé and Lichtenstein in 1942 [1]. They are most often diagnosed in childhood and early adulthood. The literature reports that ABCs comprise 1–6% of all primary benign bone tumors [2,3,4,5]. Most cases of ABCs (75–90%) are reported for patients younger than 20 years, with a slightly higher incidence for females [2,5,6]. Most common localizations are the pelvis, the metaphysis of long bones and the spine, but ABCs can also affect any other localization [2,5,7,8].

The clinical symptoms of an ABC consist of pain and swelling in the affected region and pathological fractures can be observed occasionally. In conventional radiography a relatively well-defined osteolytic, expansile lesion with possible blowout of the periosteum and a soap-bubble appearance can be found [2,5,6,8]. MRI scanning shows cystic formations with typical fluid-fluid levels due to blood sedimentation [9]. Because of the possible rapid growth with local destruction, the literature describes cases of ABCs that mimic malignant bone tumors [10,11]. Histologically, an ABC appears as a solitary, multicystic lesion, which grows rapidly and is locally destructive [4] (Figure 1).

Septae of variable thickness divide the ABC into numerous blood-filled cavities of different sizes [4,5,7,12]. ABCs have been regarded as a primary neoplasm since Panoutsakopoulos et al. [13,14] and Oliveira et al. [14,15,16] demonstrated that a recurrent chromosome aberration, t(16;17) (q22;p13), leads to a fusion gene of the entire ubiquitin-specific protease 6 (USP6 alias Tre2) coding sequence at 17p13 and the promoter region of the osteoblast cadherin 11 gene (CDH11) at 16q22 [17]. Oliveira et al. [15] identified various rearrangements involving USP6 in 69% of primary ABCs and in none of the secondary ABCs, using fluorescence in situ hybridization (FISH). Using next generation sequencing, Guseva et al. [18] and Sekoranja et al. [19] increased the number of different gene rearrangements involving USP6 observed in primary ABCs to 100%.

ABCs that are associated with a preexisting osseous lesion are defined as secondary ABCs. They represent approximately 30% of all ABCs [2,3,5]. Secondary ABCs can occur, e.g., in cases of a giant cell tumor, chondroblastoma or telangiectatic osteosarcoma [5]. The absence of rearrangements involving USP6 in secondary ABCs was confirmed by Li et al. [20]. Thus, a biopsy with histopathological examinations including cytogenetic techniques, such as FISH, is necessary in order to diagnose and to distinguish primary ABCs from secondary ABCs [10].

Different modalities of treatment are described in the literature, e.g., intralesional curettage with adjuvants, en bloc resection, embolization, percutaneous injections of polidocanol or doxycycline, injection of bone marrow, cryoablation, radiation therapy, radionuclide ablation, and the systemic use of bisphosphonates or denosumab [3,5,21,22]. The question of the optimal treatment option has not yet been clarified. The best local control rate can be achieved through an en bloc resection. Nevertheless, an en bloc resection is associated with a higher risk of complications, depending on the dimension of the resection [3]. Flont et al. [3,23] described a non-significant higher rate of postoperative pain, limb length discrepancy and strength impairment after en bloc resections than after curettage. The authors proposed an en bloc resection to be considered in cases of local recurrence [3,23]. The preferred and most commonly used therapy for primary ABCs is a less invasive, intralesional curettage. Nonetheless, an intralesional curettage comes with possible complications as well, as it is performed as open surgery. In the literature, a recurrence rate of more than 50% after intralesional curettage has been described [8]. Various adjuvants like phenol, hydrogen peroxide, defect reconstruction with PMMA, the use of a high-speed burr or electrocautery, are applied to minimize the risk of local recurrence [3,5]. A minimally invasive treatment is the percutaneous intralesional injection of different sclerosing agents. Nowadays, one of the most commonly used agents is the sclerosant polidocanol. Sclerosants damage the endothelium and start a coagulation cascade, which ends in thrombosis [24]. There are numerous authors who postulate the excellent therapeutic effect of sclerosants on ABCs [24,25,26,27]. In a study of 72 patients with primary ABCs, Rastogi et al. [25] described a successful healing rate of 97%. Puri et al. [27] confirmed the positive effect of percutaneous polidocanol instillations in the treatment of primary ABCs. 

This study analyzes the efficacy of the three most common treatment options for primary ABCs: intralesional curettage, sequential instillations of polidocanol and en bloc resection. Additionally, the value of MRI scans compared to conventional radiographs in the diagnosis, the evaluation of treatment success and the follow-up of ABCs is verified. The influence of possible risk factors such as young age, gender and localization is reviewed.

## 2. Materials and Methods

This review is a retrospective single-center study of patients with a primary ABC. The three most common treatment options are compared: intralesional curettage, sequential instillations of polidocanol and en bloc resection.

Between 2010 and 2020, 107 patients with the suspicion of an ABC underwent a biopsy. For differentiation between primary and secondary ABCs, fluorescence in situ hybridization (FISH) was used in addition to the standard histopathological examination. Thus, secondary ABCs were able to be identified and excluded from the study.

### 2.1. Radiological Classification and Measurement

For the radiological classification of the ABCs, the Capanna classification [28], the Enneking classification [29] and, in case of sequential polidocanol instillations, the classification according to Rastogi et al. [25] were applied (Table 1).

The ABCs’ volumes were measured by means of plain radiographs and/or MRI scans (Figure 2).

In analogy to Rastogi et al. [25], 10% of the measured values in plain radiographs were subtracted as a compensation for the magnification factor. For the calculation of the cyst volume, the universally valid formula for ellipsoid volumes, 4/3π (a/2) × (b/2) × (c/2), was used [30]. 

### 2.2. Curettage Group 

Until 2014, intralesional curettage was the standard treatment for every ABC in our department. Hydrogen peroxide was used as an adjuvant. After curettage, the cavity was filled with bone substitute (Actifuse^®^; Baxter Deutschland; Unterschleißheim, Germany) or PMMA (Palacos^®^; Heraeus Medical; Wehrheim, Germany).

### 2.3. Instillation Group 

Since 2014, sequential instillations of polidocanol (Aethoxysklerol^®^ 3%; Kreussler pharma; Wiesbaden, Germany) have been the standard treatment for primary ABCs in our department. The instillations were performed under general anesthesia. With a treatment interval of one month between each instillation, the procedure was repeated three times, followed by MRI scans evaluating the treatment response. The treatment was considered to be successful if (a) the clinical symptoms disappeared, (b) the MRI scans did not show fluid-fluid levels anymore and (c) the beginning of sclerosis of the cyst wall could be identified (Figure 3).

If the treatment result was unsatisfactory, further instillations were performed. After every second instillation a new MRI scan was interposed. In the case of a non-response to the instillation therapy, a conversion to intralesional curettage was performed.

### 2.4. Resection Group 

In rare cases of a primary ABC with a large soft tissue component or a massive affection of the spine, an en bloc resection was performed. In these cases, a curettage or an instillation of polidocanol were not practicable. The reconstruction of the bone defect resulted in the filling of the cavity with synthetic bone substitute, PMMA or in one case in an endoprosthetic replacement of the affected bone.

### 2.5. Follow-Up 

The treatment success was evaluated using MRI scans and plain radiographs. After instillation therapy, remaining cystic parts of the initial lesion were classified as persistent disease. After curettage, recurring cystic lesions were classified as local recurrence. The first examination took place three months after the last treatment. After six months the follow-up intervals were expanded to six months. Two years after initial treatment, the follow-up interval was expanded up to one year. 

### 2.6. Statistics 

SPSS Statistics (IBM Corp. Released 2019, Version 26.0. Armonk, NY, USA) was used for the statistical analysis. The equality of distribution among the subgroups was tested using the Kruskal–Wallis test. The Wilcoxon test was used to investigate significant changes in the volumes of the instillation subgroups’ ABCs pre- and post-intervention. To analyze the correlations of various parameters, the Chi^2^ test and the Eta coefficient were used. To compare the time until treatment requiring recurrence/persistent disease, the Kaplan–Maier curve was used. The significance analysis was performed using the Log-Rank test.

The study protocol was approved by the regional ethics committee (reference no.: 2019-592-f-S).

## 3. Results

Seventy-four patients with a diagnosis of a primary ABC were included in this study. The initial cohort consisted of 107 patients. Thirty-three patients were excluded due to a histologically diagnosed secondary ABC, due to the absence of further treatment after biopsy or short-term follow-up. Out of these 74 patients, 33 patients were male (44.6%) and 41 patients were female (55.4%, Table 2); the mean age was 16.74 (3–49) years. The mean follow-up time was 40.5 (13.1–104) months. The most common location (Table 3) among other bones was the pelvis (16; 21.1%), followed by the tibia (13; 17.1%) and the femur (11; 14.5%).

### 3.1. Cyst Volume Measurement

The mean MRI volume for each of the 26 cases in which biplanar radiographs and MRI scans were available for pre-interventional cyst volume measurement was 44.07 cm^3^ and the mean radiographic volume was 27.27 cm^3^ (Table 4). The mean difference between the two measurement techniques was 16.80 cm^3^.

### 3.2. Curettage Group 

Intralesional curettage was performed in 34 cases. The median pre-interventional cyst volume in MRI scans of the curettage group was 11.9 cm^3^ (0.5–222.6 cm^3^). After intralesional curettage the resulting cavity was filled with bone substitute in 20 cases (58.8%) and in 14 cases (41.2%) with bone cement. In 21 cases (61.8%) hydrogen peroxide was used as an adjuvant. In 18 of 34 cases (52.9%), local recurrence/persistent disease was detected. These 18 cases consisted of 16 local recurrences (47.1%) and two persistent diseases (5.9%). The mean time until local recurrence was 7.94 (2–29) months. In 15/34 cases with local recurrence (44.1%) patients needed further treatment, five patients (33.3%) were converted to instillation therapy and ten patients (66.66%) were treated with a re-curettage. Postoperative complications occurred in two cases (5.9%) in the form of a wound healing disorder. 

### 3.3. Instillation Group 

The instillation of polidocanol was performed in 32 cases. In 23/32 patients a fluoroscopic guidance for the instillation therapy was used, in nine cases a CT guided instillation. The mean number of instillations was 5.7 (1–12) and the mean volume per instillation was 6 mL (2–10 mL). This subgroup showed a mean pre-interventional volume of 46.7 cm^3^ (4.2–263.7 cm^3^) in MRI scans. After treatment a mean residual volume of 24.3 cm^3^ (0.0–451.8 cm^3^) was detected. The instillation subgroup did not present any recurrences, but 29/32 patients (90.6%) showed persistent disease. In only ten cases (34.5%) was the persistent disease classified as “requiring treatment” because of the persistence of symptoms, the presence of fluid-fluid levels (residual activity) or the enlargement of the ABC. The remaining 19 cases (65.5%) with persistent disease did not need any further treatment and were classified as “stable disease”. A total of 3/32 patients (9.4%) showed complete healing after instillation therapy. The patients with “treatment requiring persistent disease” (10; 34.5%) were treated with a curettage in nine cases (90%) and a marginal resection in one case (10%). In two cases (6.3%) a healing disorder occurred.

### 3.4. Resection Group

The resection group included eight patients who were treated with an en bloc resection. This was performed due to an ABC with a large soft tissue component in the pelvis (2; 25%), in the femur (1; 12.5%) or due to a massive manifestation in the spine (5; 62.5%). After the surgical treatment three patients (37.5%) showed treatment-associated complications, i.e., fracture (two cases), healing disorder (one case) and spondylolisthesis (one case). Two patients (25%) showed local recurrence, but only in one case was additional treatment necessary. This patient received another marginal resection of the recurred ABC.

Comparing the time until the need for further treatment, no statistically significant difference was detectable for the three treatment groups (*p* = 0.351).

## 4. Discussion

In this study, we aimed to compare the three most common treatment options for primary ABCs: intralesional curettage, sequential instillations of polidocanol and en bloc resection. We did not find statistically significant differences for sex (*p* = 0.684), for age/grouped age (*p* = 0.516/*p* = 0.975) or for pre-interventional MRI volume (*p* = 0.097). The number of cases in the curettage group (*n* = 34) and in the instillation group (*n* = 32) were comparable, whereas only eight patients were included in the en bloc resection group.

Some authors report young age and male gender to be potential risk factors for local recurrence [5,12]. However, Dormans et al. [4] did not confirm young age as risk factor. Likewise, our results cannot confirm young age (<10 years: *p* = 0.168) or male gender (*p* = 0.077) as potential risk factors for local recurrence/persistent disease with the need for treatment.

To our knowledge this is the first study using MRI scans as standard monitoring for the diagnosis, along with the evaluation of the treatment success and follow-up examinations. Comparing the pre-interventional MRI and radiographic volume for each of the 26 cases in which biplanar radiographs and MRI scans were available, it appears that measurements based on biplanar radiographs often underestimate the volume of an ABC, especially if the ABC’s wall is not clearly definable. In the majority of the published studies [25,26,27,31] the volume measurement of ABCs and the evaluation of the treatment success are based on biplanar radiographs. This leads to underestimation of the cyst volume, and volume measurement becomes even more difficult or impossible when plain radiographs in two planes are not practicable. In the present study this occurred in 38 cases (51.4%). Docquier et al. [32] described the high benefit of MRI scans for diagnosis and follow-up examinations in cases of primary ABCs. The authors reported on the k-means clustering method based on MRI scans, which allows an even more accurate measurement of the volume [32]. Thus, MRI scans are superior to biplanar radiographs in the follow-up of ABCs because of their more exact measurement regardless of the location and their higher sensitivity regarding residual cystic parts and their activity (residual fluid-fluid levels). Therefore, MRI scans should be established as standard for evaluating treatment success after instillation of polidocanol or surgical treatment of a primary ABC.

This study demonstrated a local recurrence/persistent disease rate with the need for further treatment of 44.1% (15/34) for the curettage group. Comparable studies in the literature with intralesional curettage of ABCs describe local recurrence rates of 5–50% [2,7,8,33]. In a study of 80 patients with primary ABCs, Peeters et al. [33] showed a definite lower local recurrence rate of 5% with an adjuvant cryotherapy. Through the use of a high-speed burr and phenol, Garg et al. [34] achieved a reduction of local recurrence in a small study of 12 patients and with intralesional curettage of ABCs. However, in the present study hydrogen peroxide was used as an adjuvant. Several studies [35,36] demonstrated the positive effect of hydrogen peroxide as an adjuvant in intralesional curettage of benign or intermediate primary bone tumors. The present study did not show a statistically significant difference (*p* = 0.214) between intralesional curettage with or without hydrogen peroxide regarding local recurrence.

In the instillation group a persistent disease was diagnosed in 29 of 32 cases (90.6%). In only three cases (9.4%) was complete healing seen. A total of 10/29 patients (34.5%) received further treatment due to persistent disease. The good response of primary ABCs to the instillation therapy with polidocanol demonstrated by Rastogi et al. [25] was able to be verified by the present study, showing a statistically significant reduction of the ABC’s volume after sequential instillation of polidocanol (*p* < 0.001). Rastogi et al. [25] reported a reduction of the initial volume to 50% or less in 97.2% of cases. The authors classified this significant reduction as so-called “adequate healing”. “Inadequate healing” was classified as a residual cyst volume >50% of the initial volume, which was reported in 2.8% of the cases by Rastogi et al. [25]. Referring to this, the present study showed “adequate healing” in 71.9% of the patients and “inadequate healing” in 28.1% of the cases. The more accurate cyst volume measurements of ABCs based on MRI scans and the higher mean initial cyst volume in the present study (46.72 cm^3^) compared to the study by Rastogi et al. [25] (measurement based on biplanar radiographs; 26.1 cm^3^) may be a reason for the less satisfactory result. In the present study, 8/10 patients who needed further treatment after sequential instillations had a residual volume of less than 25%. In these cases, clinical symptoms and/or a persistence of fluid-fluid levels were decisive. Therefore, it should be re-considered whether a residual cyst volume <50% after instillation therapy is a reliable parameter for enduring therapeutic success. In only two cases of the present study were the need for further treatment and the classification of “inadequate healing” according to Rastogi et al. [25] (>50% residual volume) observed to be congruent. The persistence of fluid-fluid levels as a sign of residual cyst activity seems to play a larger role in terms of clinical symptoms and the need for further treatment. Nevertheless, the present study confirms the efficacy of sequential instillations of polidocanol in the treatment of ABCs, although this did not reach the rate of “satisfactory results” presented by Rastogi et al. [25] (72% vs. 97%).

In several other studies patients with a primary ABC were successfully treated with a single instillation of polidocanol [24,25,27]. Puri et al. [27] and Rastogi et al. [25] reported a mean number of 2–3 instillations and successful healing after a single instillation. In the present study no patient was successfully treated with a single instillation. The mean number of instillations was 5.8 and the mean volume of polidocanol was 6 mL. The higher mean number of instillations may be explainable by the different types of follow-up examinations (MRI scans vs. biplanar radiographs), the differing definition of a successful treatment and a higher initial cyst volume between the above-mentioned studies and the recent study.

The modality of guidance during sclerosant instillation has been discussed in several studies before. Guibaud et al. [37] and Adamsbaum et al. [38] describe fluoroscopic guidance for easily accessible lesions. However, in challenging locations such as the spine or the pelvis, the authors recommend CT-guided instillation therapy. Batisse et al. [39] report on CT-guided injections of sclerosants in two cases of patients with ABCs of the cervical and thoracic spine. In the present study, CT-guided instillations of ABCs, mainly localized in the pelvis, were performed in 9/32 cases.

Comparing the time until the need for further treatment, no statistically significant difference was detectable for the three treatment groups (*p* = 0.351) in this study. To our knowledge, only one comparative study [26] regarding intralesional curettage and instillation of polidocanol in the treatment of ABCs has been published yet. In this prospective study with 94 patients, Varshney et al. [26] did not find any significant differences in recurrence rates between the two treatment groups. The results of the present study support the findings of Varshney et al. [26], demonstrating that percutaneous instillation of polidocanol is at least equally efficient in the treatment of primary ABCs compared to intralesional curettage. However, the results of the present study cannot confirm the outstanding results described by Rastogi et al. and Puri et al. [25,27]. The recent study complies with the request of Bavan et al. [40] for further comparative studies of different treatment options in cases of primary ABCs. Neither the recent studies in the literature, nor the present study, were able to demonstrate the superiority of one of the treatment options. Due to this, we agree with Bavan et al. [40] that further prospective multicenter studies with standardized treatment protocols are needed. Nevertheless, the treatment of primary ABCs will always be subject to individual factors as localization of the cyst, the extent of an osseous defect and the patient’s wishes.

The limitations of the recent study rely predominantly on its retrospective character and the quantity of patients in the three subgroups. The patients were not randomized. Nevertheless, the groups did not differ significantly in sex, age or pre-interventional MRI volume.

## 5. Conclusions

MRI scans are superior to biplanar radiographs in the diagnosis, therapy planning and follow-up of ABCs. Compared to intralesional curettage, this study demonstrated at least an equal efficacy of sequential polidocanol instillations for the treatment of primary ABCs. However, a series of instillations has to be planned. In a considerable number of cases, persistent disease with the need for further treatment has to be expected even if a satisfactory decrease in the initial cyst volume is observed. A conversion to intralesional curettage or en bloc resection may be necessary.

## Figures and Tables

**Figure 1 cancers-13-02362-f001:**
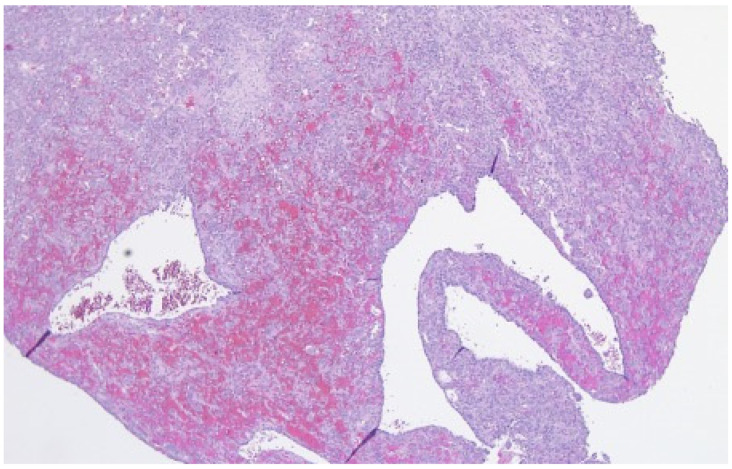
Cystic elements of an ABC.

**Figure 2 cancers-13-02362-f002:**
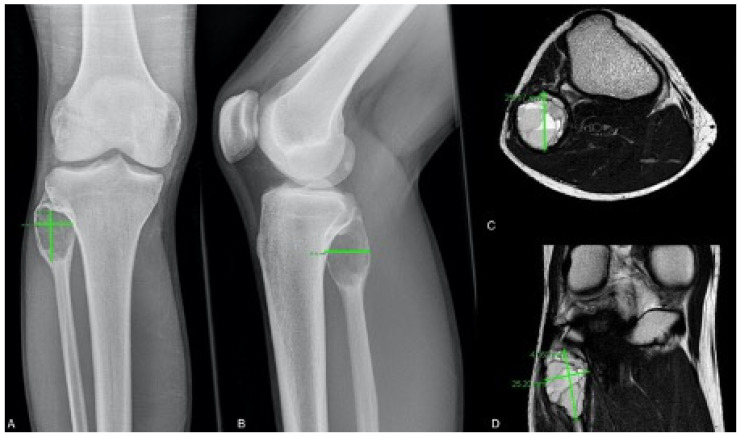
ABC of the proximal fibula of an 18-year-old patient—left: measurement of the cyst diameter in a.p. (**A**) and lateral (**B**) radiograph and in transversal (**C**) and coronar (**D**) MRI scan.

**Figure 3 cancers-13-02362-f003:**
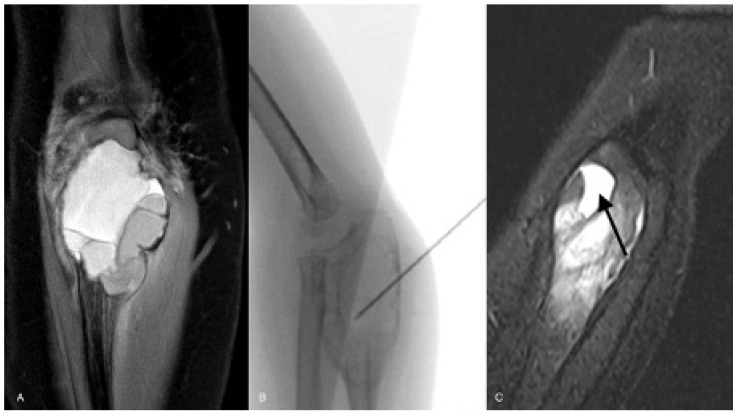
ABC of the proximal ulna of a 3-year-old girl—initial MRI scan (**A**); fluoroscopy of instillation of polidocanol (**B**); healing grade I according to Rastogi et al. with complete resolution of fluid-fluid levels ((**C**): arrow).

**Table 1 cancers-13-02362-t001:** Overview of radiological classifications.

Radiological Classification		Instillation Group	Curettage Group	En Bloc Resection Group	Over All
**Enneking classification**	still	1	1	0	2
	active	8	15	0	23
	aggressive	23	18	8	49
**Capanna classification**	I	1	1	0	2
	II	5	4	0	9
	III	7	8	1	16
	IV	1	0	0	1
	V	2	6	0	8
	not applicable	16	15	7	38
**Rastogi classification**	I	20	-	-	20
	II	3	-	-	3
	II	3	-	-	3
	IV	6	-	-	6

**Table 2 cancers-13-02362-t002:** Overview of the study cohort.

Parameter		Instillation Group	Curettage Group	En Bloc Resection Group
Number		32	34	8
Gender (male:female)		13:19	17:17	3:5
Mean age (years)		16.88	17.15	15.5
Age groups	≤10 years	5	5	1
	>10 years	27	29	7
Mean Pre-interventional cyst volume (cm^3^)		46.72	23.66	48.01
Mean number of instillations		5.74	-	-
Mean volume of instillation (mL)		6.02	-	-
Mean residual cyst volume (cm^3^)		24.3	-	-
Additional therapy	embolization	8	2	2
	intralesional cortisone injection	0	1	0
	Denosumab therapy	0	0	1
Local recurrence	no need for treatment	0	3	1
	in need of treatment	0	13	1
	over all	0	16	2
Persistent disease	no need for treatment	19	0	2
	in need of treatment	10	2	0
	over all	29	2	2
Complications	pre-interventional	5	3	3
	post-interventional	2	2	3
Mean follow-up (months)		36.12	41.36	54.73

**Table 3 cancers-13-02362-t003:** Overview of the affected bones.

Localization	Instillation Group	Curettage Group	En Bloc Resection Group	Overall
Humerus	5	4	0	9
Ulna	2	0	0	2
Hand	1	0	0	1
Femur	2	8	1	11
Tibia	7	6	0	13
Fibula	0	1	0	1
Foot	4	5	0	9
Clavicula	3	2	0	5
Scapula	1	1	0	2
Spine	0	1	5	6
Pelvis	7	6	2	15
Total	32	34	8	74

**Table 4 cancers-13-02362-t004:** Comparison of patients’ cyst volumes in MRI scans, conventional radiographs and the localization.

Patient ID	MRI Volume (cm^3^)	Radiographic Volume (cm^3^)	Difference	Localization
1.	18.5	8.55	9.95	Tibia
2.	38.67	22.45	16.22	Humerus
3.	4.96	3.26	1.7	Tibia
10.	17.64	4.10	13.57	Foot
11.	77.28	63.46	13.82	Tibia
12.	34.35	10.64	23.71	Femur
13.	31.85	6.20	25.65	Foot
16.	54.83	38.61	16.22	Humerus
19.	13.87	9.59	4.28	Ulna
21.	105.33	55.53	49.80	Humerus
26.	29.41	17.96	11.53	Tibia
27.	11.24	6.27	4.97	Foot
31.	6.67	1.25	5.42	Ulna
32.	263.71	184.58	79.13	Femur
39.	21.59	3.86	17.73	Humerus
43.	2.60	1.49	1.11	Femur
45.	31.77	17.43	14.34	Humerus
50.	1.12	0.93	0.17	Foot
51.	16.47	10.45	6.02	Fibula
52.	10.82	17.09	−6.27	Humerus
54.	56.50	26.80	29.70	Femur
55.	35.08	8.25	26.83	Tibia
67.	16.56	17.47	−0.91	Tibia
74.	11.87	4.13	7.74	Spine
75.	5.42	1.48	3.94	Spine
76.	227.64	167.19	60.45	Femur
Mean	44.07	27.27	16.8	

## Data Availability

The data presented in this study are available on request from the corresponding author. The data are not publicly available due to legal, ethical and privacy issues.

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
