# Peer review of "Primary Aneurysmal Bone Cyst and Its Recent Treatment Options: A Comparative Review of 74 Cases"

_cancers, 2021, doi:10.3390/cancers13102362_

Round 1

Reviewer 1 Report

Dear Editor:

   Thank you for the opportunity to review the manuscript by Deventer et al entitled: “ Primary aneurysmal bone cyst and its recent treatment options: 2 a comparative review of 74 cases”.

  1. retrospective and prospective review is very confusing. I am very unclear from how this is written what the prospective part of this is.   To be prospective, this would have to be a clinical trial that patients consent to.  If this is true, this would be reported differently with trial consort diagrams and clinical trial outcomes..  Please either explain what I am missing, or change it to a retrospective review.
  2. Otherwise, this is well written as is

Author Response

Dear editor,

dear reviewer,

thank you for your encouraging and helpful comment. By revising the manuscript I tried to address all aspects you mentioned in your mail/review.

(Point 1): In the abstract and methods section I classified the study as retrospective as you proposed.

Thank you again for your help and consideration of this manuscript.

Please address all correspondence concerning this manuscript to me at niklas.deventer@ukmuenster.de

Sincerely,

Dr. med. N. Deventer

Reviewer 2 Report

Dear Authors,

I feel that your research is of interest for the readers,

I suggest You some improvements: 

  • In 'conclusion' section you stated that MRI is superior to XR. Please, if a goal of your research is to compare imaging modalities add clearly this concern also in the aim of the study (both abstract and main text).
  • In 'methods' section, the formula used is not for spherical objects but for ellipsoid. Please correct this and provide one or more valid references.
  • The reference [3] is not suitable for fluid-fluid levels radiological description. I suggest this more specific: "Fluid-Fluid Levels in Aneurysmal Bone Cysts. J Pediatr. 2019 Jan;204:317. doi: 10.1016/j.jpeds.2018.08.081. Epub 2018 Sep 27. PMID: 30270163."
  • In difficult lesions' locations such as the spine, the fluoroscopic guidence should be insufficient while the CT one may be advatageous for instillation. Please discuss briefly the different radiological guidances with some references in 'discussion' section.

Author Response

Dear editor,

dear reviewer,

thank you for your encouraging and helpful comments. By revising the manuscript I tried to address all aspects you mentioned in your mail/review.

  • (Point 1): As you proposed I expanded the aims of the study (in the abstract and main text) concerning the comparison between the imaging modalities (MRI vs. conventional radiographs) in diagnosis, treatment evaluation and follow-up of ABCs.
  • (Point 2): I replaced the term spherical by ellipsoid for cyst volume calculation and added a suitable reference (Gobel, V.; Jurgens, H.; Etspuler, G.; Kemperdick, H.; Jungblut, R.M.; Stienen, U.; Gobel, U. Prognostic significance of tumor volume in localized Ewing's sarcoma of bone in children and adolescents. J Cancer Res Clin Oncol 1987, 113, 187-191, doi:10.1007/BF00391442.). According to the reference the formula was adjusted to 4/3 π(a/2) x (b/2) x (c/2).
  • (Point 3): I replaced the reference concerning fluid-fluid-levels in ABCs by the reference you mentioned ("Fluid-Fluid Levels in Aneurysmal Bone Cysts. J Pediatr. 2019 Jan;204:317. doi: 10.1016/j.jpeds.2018.08.081. Epub 2018 Sep 27. PMID: 30270163.")
  • (Point 4): As you mentioned, I discussed the use of fluoroscopic and CT guidance for instillations of ABCs in difficult locations and added suitable references.

Thank you again for your help and consideration of this manuscript.

Please address all correspondence concerning this manuscript to me at niklas.deventer@ukmuenster.de

Sincerely,

Dr. med. N. Deventer

Round 2

Reviewer 2 Report

Dear Authors,

I appreciate the new version of the manuscript.

I feel that is now suitable for publication.